# NMR analysis of nucleotide π-stacking in prebiotically relevant crowded environment

Niraja V. Bapat[1,3], Harshad Paithankar [2,3], Jeetender Chugh [1,2✉] & Sudha Rajamani [1✉]

The prebiotic soup of a putative 'RNA World' would have been replete with a plethora of molecules resulting from complex chemical syntheses and exogeneous delivery. The presence of background molecules could lead to molecular crowding, potentially affecting the course of the reactions facilitated therein. Using NMR spectroscopy, we have analyzed the effect of crowding on the stacking ability of RNA monomers. Our findings corroborate that the purines stack more efficiently than the pyrimidine ribonucleotides. This competence is further enhanced in the presence of a crowding agent. This enhanced stacking could result in greater sequestration of the purine monomers, putting their ready availability for relevant nonenzymatic reactions into question. Thus, this study demonstrates the need for systematic characterization of molecular crowding in the context of prebiotically pertinent processes. Unraveling such phenomena is essential for our understanding of the transition from abiotic to biotic, during the origin of life.

[1] Department of Biology, Indian Institute of Science Education and Research (IISER), Dr. Homi Bhabha Road, Pashan, Pune 411008, India. [2] Department of Chemistry, Indian Institute of Science Education and Research (IISER), Dr. Homi Bhabha Road, Pashan, Pune 411008, India. [3]These authors contributed equally: Niraja V. Bapat, Harshad Paithankar. ✉email: cjeet@iiserpune.ac.in; srajamani@iiserpune.ac.in

Chemical origin of life posits that the prebiotic soup would have facilitated important chemical reactions, which are thought to have led to the emergence of life on early Earth. This composite solution would have been replete with different kinds of molecules with varied reactivity and interactions that might have impinged on the processes leading to the emergence of the first cell-like entities. The heterogenous nature of the soup would have been partly due to the highly diverse range of products that would have resulted from various prebiotically relevant syntheses. It is well known that the outcome of most of such reactions is a mixture of products. For example, the formose synthesis reaction that results in a mixture of four to six carbon containing sugar molecules, never exclusively yields only one kind of sugar[1]. Along with D-ribose that is found in RNA molecules, this reaction also yields a plethora of other sugars such as erythrose, xylose, arabinose, glucose, mannose, etc. The Fischer-Tropsch-type synthesis pathway, which is considered as a prebiotically viable means of synthesizing longer carbon chain compounds from carbon monoxide and hydrogen, is also known to yield a complex mixture of alkanols, alkanoic acids, alkenes, and alkanes, containing anywhere from 2 to over 30 carbon moieties[2]. Similarly, only a fraction of the resultant product from Oró's synthesis is constituted of the canonical adenine nucleobase[3]. On the other hand, the meteorites that would have fallen on the prebiotic Earth would have been comprised of a mixture of compounds; the analysis of Murchison meteoritic samples has confirmed the presence of an abundance of sugar-related compounds and organic molecules[4,5].

These examples clearly indicate that the early Earth would have been rife with different kinds of molecules that would have been present concomitantly in the prebiotic soup. Therefore, any prebiotically pertinent reaction could not have taken place in isolation and under buffered conditions, unless completely secluded by some relevant mechanism. The existence of various types of molecular entities (cosolutes or background molecules) would have resulted in a complex prebiotic milieu that, consequently, would have facilitated phenomenon like molecular crowding. Crowding, for example, has been shown to affect, both, the kinetics and equilibrium of several biochemical reactions[6]. Macromolecular crowding has also been shown to impact protein folding, stability, and rates of the enzymatic reactions that they catalyze[7,8]. Significantly, it has also been demonstrated to influence reactions involving nucleic acids[9]. In the presence of water-soluble crowding agents, the thermal stability of DNA duplexes (ranging from 8 to 30 mer) is known to be dependent on the DNA length and the size of the cosolutes[10]. A few studies have also delineated the effect of molecular crowding on prebiotically relevant molecules like ribozymes. For example, catalytic RNAs are known to get stabilized under crowded conditions[11,12] and show enhanced activity in the presence of cosolutes[11–14].

In addition to the direct interactions, there is another conceivable way of explaining the effect that crowding agents could exert on macromolecule-based reactions. Crowding could potentially alter molecular diffusion that is precipitated by the presence of bulky cosolute polymers. Crowding agents are known to reduce the diffusion coefficient ($D$) of, both, small and large molecules[6]. Diffusion of the monomers would have played an important role in polymer formation during the origin of life. Interestingly, restricting monomer diffusion using certain methods such as surface adsorption on clay mineral[15], entrapment of monomers in dried lipid films[16,17], and crowding of monomers in the liquid channels within eutectic ice phases[18,19] etc., has been demonstrated to enhance the yields of nonenzymatic polymerization reactions.

In this study, we analyze the changes in the diffusion and, thereby, the stacking properties of RNA monomers in aqueous solution upon addition of a molecular crowding agent. To our knowledge, this study is possibly the first of its kind wherein RNA monomer diffusion has been evaluated, especially in the prebiotic context. Polyethylene glycol (PEG) and dextran, which are bulky water-soluble inert polymers that are used to simulate crowded conditions in several biochemical studies[8], have been used as molecular crowding agents in our reactions. Nuclear Magnetic Resonance (NMR) spectroscopy has been used as the main analytical tool as it is a noninvasive technique and does not require any tagging of the molecules in question. When nucleotides stack, they would get clustered and their effective molecular size would increase, and this is evident from both our DOSY (which probes translational diffusion process) and $T_1$ NMR data (which probes translational and rotational diffusion processes). Our observations indicate that the purine monomers stack to a greater extent than pyrimidine monomers, under crowded conditions. This could affect their availability for participating in prebiotically relevant reactions such as nonenzymatic polymerization and replication of nucleic acids. In a relevant study undertaken by us, we had reported that PEG and DLPC lipid vesicles, when used as cosolutes, reduced the rate of purine-based addition reactions in template-directed copying of RNA, thus, resulting in an overall reduction in the fidelity of the copying of information in this RNA-based system[20]. The present study and the aforementioned one highlight the possibility of how such prebiotically pertinent scenarios could have fundamentally impacted events that would have facilitated the transition from chemistry to biology on the early Earth.

## Results

**Nucleotide concentration affects translational diffusion.** To begin with, the DOSY NMR data were recorded for the nucleotide samples in the absence of PEG. The $D$ obtained for these samples are listed in Supplementary Table 1. Concentration dependent consistent decrease in $D$ was observed for all four nucleotides, suggesting slowing down of the translational diffusion in the solution (Fig. 1). However, the DOSY data recorded for nucleotide samples in the presence of PEG did not show any clear trend. This could arise since diffusion processes are governed primarily by the high viscosity of the solution in the presence of PEG. Hence, $^{13}C$-$T_1$ relaxation experiment was subsequently used as a tool to compare the relative sizes of the molecules in the presence and absence of PEG and dextran.

**Nucleotide concentration affects rotational correlation time.** The $^{13}C$-$T_1$ relaxation data were recorded for all four 5'-nucleoside monophosphates (5'-NMPs) at three different concentrations and at two temperatures viz. 10 and 25 °C. The theoretical $T_1$ relaxation time follows a curve (Eq. 1) passing through a minimum as the molecular size increases. Figure 2 shows $^{13}C$-$T_1$ relaxation time plotted against the rotational correlation time of the molecule, which is a determinant of the molecular size. The $T_1$ values are obtained through simulations using following equations:

$$\frac{1}{T_1} = \left(\frac{\delta^{CH}}{4}\right)^2 \left[J(\omega_H - \omega_C) + 3J(\omega_C) + 6J(\omega_H + \omega_C)\right] \\ + \frac{3}{4}\left(\omega^C \sigma_{zz}^C\right)^2 J(\omega_C), \tag{1}$$

$$\delta^{CH} = -2\frac{\mu_0 \gamma_H \gamma_C \hbar}{4\pi r_{CH}^3}, \tag{2}$$

$$J(\omega) = \frac{2}{5}\frac{\tau_C}{1 + (\omega\tau_C)^2}, \tag{3}$$

where $\delta^{CH}$ = anisotropy of the dipolar coupling between C and H;

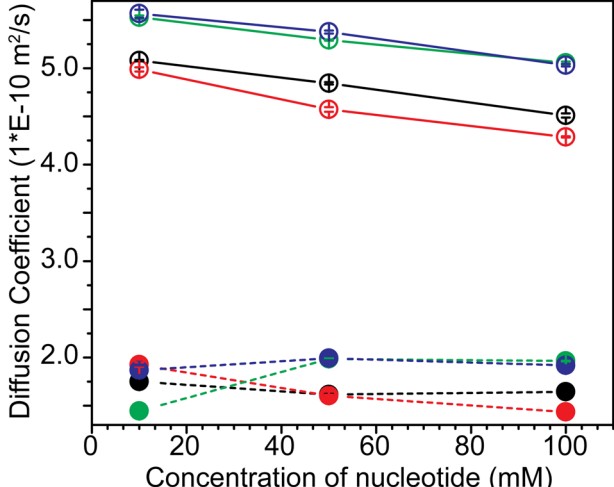

**Fig. 1 DOSY NMR measurements for nucleotide diffusion coefficients.**
The diffusion coefficients were measured at different concentrations, in the absence (indicated by empty circles connected by a solid line) and in the presence of PEG (indicated by filled circles, connected by a dashed line), for all four nucleotides viz. 5′-AMP (indicated in black), 5′-GMP (indicated in red), 5′-CMP (indicated in green), and 5′-UMP (indicated in blue) (also see Supplementary Table 1). The x-axis indicates three different concentrations at which the data were recorded. Errors in the diffusion coefficients are shown by the bars and are of the order of 1%. The error bars indicate the standard error obtained by data fitting to Eq. (4).

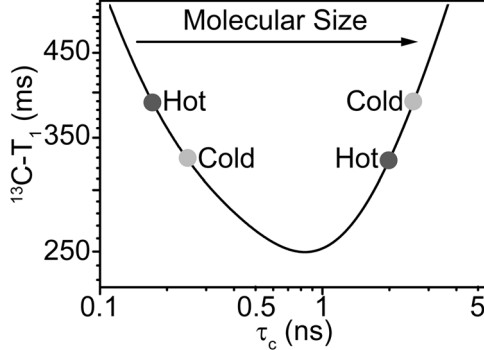

**Fig. 2 $^{13}C$-$T_1$ relaxation time as a function of molecular size shown as a temperature-dependent quantity.** The log–log plot of $^{13}C$-$T_1$ relaxation time as a function of molecular size, characterized by rotation correlation time ($\tau_c$), is shown as a temperature-dependent quantity. The curve has been simulated using Eqs. (1–3) to represent data measured at NMR spectrometer with magnetic field strength of 14.1 T, $\omega_H = 14.1 \times \gamma_H$, $\omega_C = 14.1 \times \gamma_C$, and chemical shielding tensor ($\sigma_{zz}$) = 0 ppm.

$\mu_0$ = vacuum permittivity = $4\pi \times 10^{-7}$ N A$^{-2}$;
$\gamma_C$ and $\gamma_H$ = gyromagnetic ratio of $^{13}C$ and $^1H$ (6.728 × $10^7$ T$^{-1}$ s$^{-1}$ and 2.675 × $10^8$ T$^{-1}$ s$^{-1}$, respectively);
$\hbar$ = reduced plank constant (1.05457 × $10^{-34}$ J s);
$r_{CH}$ = distance between two nuclei C and H (1.1 Å);
$J(\omega)$ = spectral density at frequency $\omega$;
$\omega_C$ and $\omega_H$ = resonance frequency of $^{13}C$ and $^1H$;
$\sigma_{zz}$ = Chemical shielding tensor; and
$\tau_C$ = Rotational correlation time of the molecule.
For the molecules with size characterized by left of the $T_1$ minimum, $T_1$ time increases with an increase in sample temperature, while for macromolecules which have size greater than that characterized by $T_1$ minimum, relaxation time decreases with an increase in temperature[21].

The comparison of $T_1$ relaxation times for all these samples showed an increase with increase in the temperature from 10 to 25 °C (Fig. 3 and Supplementary Table 2). This suggested that the nucleotides behaved as small molecules (characterized by the left of the $T_1$ minimum in Fig. 2) under our analytical conditions. As the concentration of nucleotides was increased from 10 to 100 mM, a significant decrease in $T_1$ relaxation time ($\Delta T_1 \sim$ 150 ms at 25 °C and $\Delta T_1 \sim$ 100 ms at 10 °C) was observed for purine monomers (adenosine 5′-monophosphate (5′-AMP) and guanosine 5′-monophosphate (5′-GMP)). No such significant change was observed in the case of pyrimidine monomers (uridine 5′-monophosphate (5′-UMP) and cytidine 5′-monophosphate (5′-CMP)). The decrease in $T_1$ for purine monomers with an increase in the nucleotide concentration suggested an overall increase in the rotational correlation time pointing towards increased stacking.

**Molecular crowding slows down rotational correlation time.** The recorded $T_1$ values were, in general, lower in the presence of PEG and dextran as compared with those obtained in their absence due to an increase in the viscosity of the solution. In presence of either of the crowding agents, the $T_1$ relaxation time showed a similar increase (as was seen in their absence) from cold (10 °C) to hot (25 °C) measurement condition, implying that the overall molecular size is characterized by left of the $T_1$ minimum as described in Fig. 2. This suggested that the nucleotides behaved like small molecules even in the presence of the crowding agent. However, upon increasing the nucleotide concentration from 10 to 100 mM, the difference in the $T_1$ values of the hot and cold conditions was found to decrease for purine monomers (except for 10 mM 5′-GMP) (Fig. 3, Supplementary Fig. 2, and Supplementary Table 2), while it was invariant for pyrimidine monomers (was observed in case of both the crowding agents). This suggested an increase in the size of the nucleotide cluster/pseudo-oligomers in the presence of PEG or dextran.

For, 10 mM of 5′-GMP, the recorded $T_1$ relaxation time was higher at 10 °C than at 25 °C, in the presence of PEG (Supplementary Table 2). This meant that in the presence of PEG, at lower concentration, 5′-GMP acted as a large molecule as estimated by this $T_1$ relaxation data. However, at higher concentrations of 40 and 100 mM of 5′-GMP, the recorded $T_1$ values in the presence of PEG were found to be lower at 10 °C (Supplementary Table 2). Upon $^1H$ NMR analysis, the peak corresponding to the presence of G-quadruplex was observed at 40 mM 5′-GMP and was absent at 10 mM 5′-GMP (Supplementary Fig. 1). This observation supported the $T_1$ data by confirming the formation of compact G-quadruplex-like structures at higher GMP concentration as compared with the looser stacks potentially present at lower GMP concentrations. Similar trend of $T_1$ rates was also observed in the case of dextran. Nonetheless, the G-quadruplex peak was not seen in the $^1H$ NMR analysis of 40 mM GMP in dextran. This could possibly be due to less compact packing of the pseudo-oligomers of 5′-GMP in the presence of dextran, as is indicated by a reduced increase in the corresponding $T_1$ rates from 10 to 25 °C.

## Discussion

The heterogeneous nature of the prebiotic milieu would have resulted in interesting phenomenon such as molecular crowding and related events. The presence of molecular crowding agents is known to affect the $D$ of the solutes. If the $D$ is reduced, molecules would take relatively longer to travel the same distance, thus, resulting in reduced chances of encountering neighboring molecules or their interacting partners. Hence, if the reaction is diffusion limited, its rate would decrease in the presence of cosolute

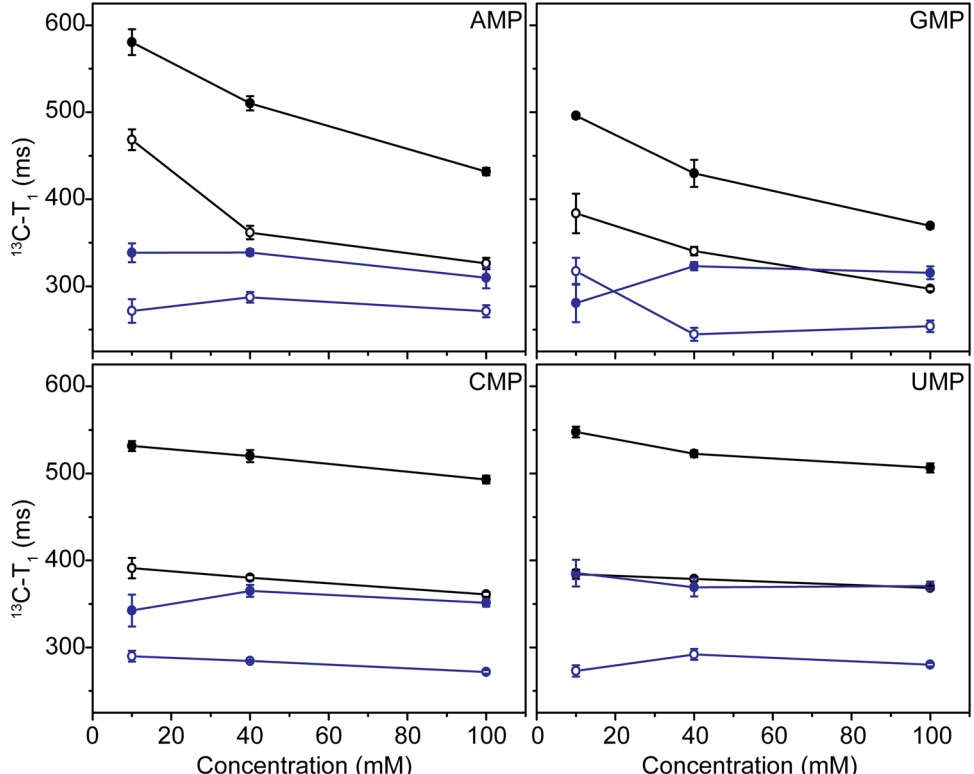

**Fig. 3 $^{13}$C-$T_1$ relaxation time data for different concentrations of nucleotides.** The data for all four nucleotides were recorded either in the absence of crowding agent (Black) or in the presence of PEG (blue), at 25 °C (indicated by filled circle) and 10 °C (indicated by empty circle), respectively. The nucleotide detail is mentioned in the top right corner of each of the plot. The error bars are obtained as a fit error to the monoexponential decay Eq. (5), using repeat point and Monte Carlo simulations.

polymers[6]. It has been previously reported that the presence of PEG and lipid vesicles affect the rate and fidelity of template-directed nonenzymatic RNA copying reactions[20]. The results from this study underscored the importance of accounting for the presence of background molecules in prebiotically relevant reactions. This is especially pertinent as their presence could have directly impacted the kinetics of nonenzymatic oligomerization and replication processes, thereby affecting the origin and composition of the resultant informational polymers.

In the present study, we have analyzed the effect of molecular crowding on the diffusion of prebiotically relevant molecules viz. RNA monomers. RNA has been argued to be the first biomolecule to have arisen during the origin of life due to its dual ability of carrying genetic information and performing catalysis[22]. NMR analysis was used to probe the effect of crowding agents on ribonucleotide stacking and thereby its effect on nucleotide diffusion under our analytical conditions. Along with being a non-invasive method, NMR also bypasses the need to tag the nucleotides to track their movement. This is hugely advantageous as the attachment of bulky fluorescent tags would increase the effective nucleotide size, thus, likely affecting their stacking properties and thereby their diffusion.

DOSY NMR was the initial method of choice for this analysis as one can get a direct estimation of $D$ for molecules under different solution conditions. In the absence of PEG in the solution, the $D$ showed a decrease with a concurrent increase in the concentration for all the four canonical 5′-NMPs, suggesting an increase in the size of the stack of monomers with an increase in concentration. The $D$ observed in the presence of PEG were lower in general; however, concentration dependent decrease was not apparent. This might be due to the overall effect of PEG on the viscosity of the solution. This could level out the effect of

nucleotide stacking on $D$ such that no observable differences were detected in the D, even at different concentrations. Importantly, spontaneous formation of columnar liquid crystals has been recently reported for DNA and RNA monomers at high concentrations and low temperature[23].

Since, the $D$ calculated from DOSY could not provide insights into the differential size estimation for the various nucleotide concentrations studied in the presence of PEG, $T_1$ relaxation time was subsequently recorded. $T_1$ relaxation, which is also known as spin-lattice relaxation time, is a temperature-dependent function of the molecule and hence can be used to estimate its size based on the rotational properties. In the absence of crowding agent, an increase in the $T_1$ value was observed for the nucleotides for a concurrent increase in the temperature from 10 to 25 °C. This suggested that the nucleotides act as small molecules under this experimental setup[21] and lie on the left of the $T_1$ minimum at all concentrations. However, a notable concentration dependent decrease in $T_1$ was observed only for purine monomers at both the temperatures. This stems from better $\pi$-stacking properties exhibited by the purines, which might lead to the formation of pseudo-oligomers ultimately resulting in larger sized ensembles at higher concentrations. Pertinently, the $D$ has been shown to be dependent on the number of nucleotides that are present in an ssRNA[24]. Therefore, larger molecular size would result in a concurrent decrease in diffusion such that the stacked purine monomers might be less available for the diffusion-limited reactions as against what was observed for the pyrimidine monomers. Significantly, this effect could further get enhanced at higher concentrations of the nucleotides (Fig. 3).

In the presence of PEG, the $T_1$ relaxation time was observed to have decreased in general for all the nucleotides and at all concentrations. This could be primarily due to the increase in the

viscosity of the solution in the presence of bulky PEG polymers[25]. When the data were analyzed closely, it was noted that the overall decrease in the $T_1$ values at 25 °C, compared with those observed at 10 °C, was lower for the purine nucleotides as compared with those obtained for pyrimidine nucleotides (Supplementary Table 2). For example, the $T_1$ for 100 mM 5′-AMP at 25 °C was found to be around 310 ms and that at 10 °C was ~270 ms (the difference being that of ~40 ms). On the other hand, the $T_1$ for 100 mM 5′-UMP at 25 °C was found to be 370 ms and that at 10 °C was 280 ms (i.e., the difference of ~90 ms). The reduction in the $T_1$ value from higher temperature to the lower temperature would be less for a molecule/molecular cluster that is comparatively of larger size (Fig. 2)[21]. Thus, this variation in the difference is suggestive of higher sized molecular clusters/pseudo-oligomers for purine monomers in the presence of PEG, than that for pyrimidine monomers. The molecular clusters/pseudo-oligomers formed by purines are concentration independent in the concentration range tested, as evidenced by no significant change in the increase in $T_1$ times from 10 to 25 °C (Fig. 3 and Supplementary Table 2). Thus, PEG might facilitate higher degree of stacking for purine monomers, resulting in the sequestration of these monomers into pseudo-oligomeric structures. Similar trend of $T_1$ relaxation rates were also observed in reactions with dextran as the crowding agent (Supplementary Fig. 2 and Supplementary Table 2). This suggests that the higher degree of stacking observed for purines is not limited only to PEG and could be a general effect stemming from the presence of crowding agents in the reactions. Nonetheless, the possibility of H-bond mediated association in addition to stacking may not be ruled out completely.

Another peculiar thing that was observed from the relaxation data recorded in the presence of PEG was the $T_1$ values for 5′-GMP. This is interesting because molecular crowding induced by PEG is known to promote formation of noncanonical G-quadruplex structures that result from the formation of non-Watson-Crick interactions between guanine[26]. We, therefore, suspected that in the presence of PEG, 5′-GMP could be forming G-quadruplex-like structures at higher concentrations resulting in the compaction of the nucleotide clusters, and thus, yielding a $T_1$ relaxation time trend characteristic of a small molecule. The formation of hydrogen bonds in G-quadruplexes results in the stabilization of exchangeable hydrogen, which can be observed as a peak at ~10–12 ppm on $^1H$ NMR. Our $^1H$ NMR analysis confirmed the formation of G-quadruplex structure at higher concentration of 5′-GMP in presence of PEG.

The effect of molecular crowding on π-stacking might result in the formation of pseudo-oligomers wherein the nucleotides cluster, potentially resulting in a noncovalently linked polymer-like structure. This would increase their effective molecular size, slowing down their diffusion, which is evident from both our DOSY (which probes translational diffusion process) and $T_1$ NMR data (which probe translational and rotational diffusion processes). Further, the $T_1$ data also indicate that this stacking tendency increases in the presence of crowding agents, especially in the case of purines. It can be said that both H-bonding and stacking interactions could potentially be involved in the aforesaid phenomenon, and these might be in addition to a possible contribution from solvent organization (to be discerned in future studies). Consequently, this would result in the sequestration of a higher percentage of purine monomers in such aggregates, affecting their availability for prebiotically relevant processes like nonenzymatic template-directed replication reactions that we reported in our 2015 study[20]. The results from the present study explain, in part, the previously observed reduction in the rate of purine-based cognate nonenzymatic addition reactions, particularly in the presence of PEG. The interaction that we propose here is based on $D$ and transverse relaxation time data, which reflect

on the potential stacking interactions between the nucleotides. Intrinsically, this is a difficult interaction to probe by mass determining techniques as the steps involved in demonstrating this is nontrivial and would adversely affect the stacks that are very delicate to begin with (as they possibly exist without any support from covalent or other noncovalent interactions). Nonetheless, the aforementioned orthogonal approaches involving both DOSY and $T_1$ relaxation studies, ended up giving similar results highlighting our primary observation detailed above, thereby strongly indicating the presence of large soluble species in solution.

Significantly, these observations demonstrate the importance of accounting for heterogeneity in the prebiotic soup and understanding how it would impinge on prebiotically pertinent nonenzymatic processes. The presence of relevant cosolute molecules should, therefore, be factored in prebiotic reaction schemes while characterizing pertinent nonenzymatic copying reactions. This is crucial as the interactions facilitated by these cosolutes could directly impinge on the propagation of genetic information during the origin of life on Earth. In conclusion, the work detailed in here underscores how heterogeneous mixtures and the emergent phenomena that they facilitate, could have direct consequences for characterization of relevant prebiotic processes.

## Methods

**Chemicals**. The disodium salts of all four 5′-NMPs, viz. 5′-AMP, 5′-GMP, 5′-UMP, and 5′-CMP (all purity ≥ 98%), were purchased from Sigma-Aldrich (Bangalore, India) and used without any further purification. Analytical grade PEG 8000, dextran (average mol. wt. of 9,000- 11,000), and deuterated water ($D_2O$) were also purchased from Sigma-Aldrich.

**Sample preparation**. The nucleotide stocks were prepared in nanopure water and the concentrations were estimated using UV spectroscopy (UV-1800 UV/Vis spectrophotometer, Shimadzu Corp., Japan). Stock solutions with 50% w/v and 40% w/v were prepared for PEG 8000 and dextran, respectively, by dissolving the required amount of powder in nanopure water. Three different concentrations of the nucleotides viz. 10, 50, and 100 mM, were used to record the DOSY NMR data. For $T_1$ relaxation time measurements, the data were recorded for nucleotide concentrations of 10, 40, and 100 mM. The final concentration of PEG 8000 and dextran in the analyzed samples was maintained at 18%. Overall, 300 μl of 1.1 times concentrated samples was first prepared and 10% $D_2O$ was subsequently added to these samples for field locking before recording the NMR data.

**NMR data acquisition**. All the DOSY NMR and $T_1$ relaxation experiments were recorded on Bruker 600 MHz NMR spectrometer, furnished with quadruple ($^1H$/$^{13}C$/$^{15}N$/$^{31}P$) resonance cryoprobe equipped with X-, Y-, and Z-gradient; and dual receiver operating. DOSY experiments were recorded for the nucleotides in the presence and absence of PEG 8000. For each sample, the diffusion time and the gradient length was optimized to get 5–10% residual signal at 95% of the maximum gradient strength (42.58 G/cm). Sixteen data points were recorded with strength of the gradient ranging between 2 and 95%. The diffusion times (Δ), and the durations for which the gradient pulse (δ) was applied for all the analyzed samples, are listed in Supplementary Table 3 in the supplementary data section.

For $^{13}C$-$T_1$ relaxation time measurement, $^{13}C$ signals were excited and detected on the attached protons via polarization transfer. The $^{13}C$ data were recorded by providing six inversion recovery delays (five delay points and a repeat point for error estimation) in the range of 10–800 ms. The recovery delays were chosen to get 70% reduction in the signal intensity at the longest delay. For all the samples (i.e., control samples as well as those containing 18% PEG 8000 or dextran), the $T_1$ relaxation data were recorded at two temperatures viz. 10 °C (cold) and 25 °C (hot).

**Data analysis**. The collected DOSY data were processed using SimFit algorithm as explained in standard Bruker DOSY data processing manual (Kerssebaum, R. DOSY and Diffusion by NMR. In User Guide for XWinNMR 3.5, Version 1.0; Bruker Buospin GmbH: Rheinstetten, Germany, 2002). The intensities extracted by SimFit algorithm were fit using a two-parameter monoexponential Eq. (4) in OriginPro 8.5.0 (OriginLab, Northampton, MA, USA) to get the $D$ for the solute molecule in the solvent that has been used in the study:

$$I = I_0 \exp\left[-D\gamma^2 g^2\delta^2\left(\Delta - \frac{\delta}{3}\right)\right], \qquad (4)$$

where $I$ is the observed intensity, $I_0$ is the reference or unattenuated intensity, $D$ the

diffusion coefficient, $\gamma$ the gyromagnetic magnetic ratio of the observed nucleus, $g$ the gradient strength, $\delta$ the length of the gradient pulse, and $\Delta$ the diffusion time.

The $^{13}C$-$T_1$ relaxation data were processed and extracted as separate 1D corresponding to each recovery delay time, $t$. Peak picking was done in topspin3.2 (Bruker Biospin, GmbH: Rheinstetten, Germany). The intensity data were then fitted using Mathematica v5.2 (Wolfram Research, Inc., Champaign, IL, USA) script[27] to the following monoexponential decay function, to get the longitudinal relaxation time constant ($T_1$):

$$I = I_0 \exp\left(-t/T_1\right). \quad (5)$$

The error in the $T_1$ time was obtained as fit error the to the above Eq. (5) using both Monte Carlo simulations and repeat relaxation delay point from the Mathematica script.

## Data availability

The NMR data that support the findings of this study are available in Mendeley Data with the identifier https://doi.org/10.17632/xbvrxxzphj.1.

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

## Acknowledgements
The authors wish to acknowledge the High Field NMR facility at IISER—Pune (co-funded by DST-FIST and IISER, Pune). J.C. acknowledges Department of Biotechnology, Govt. of India [BT/PR24185/BRB/10/1605/2017], and the Science and Engineering Research Board (SERB), Govt. of India [EMR/2015/001966] for extramural funding. S.R. acknowledges Department of Biotechnology, Govt. of India [BT/PR19201/BRB/10/1532/2016] for extramural funding. N.V.B. acknowledges the research fellowship received from CSIR, Govt. of India.

## Author contributions
N.V.B., J.C., and S.R. designed the experiments based on N.V.B and S.R.'s observations reported in a previous article. N.V.B. prepared the samples. H.P. recorded the NMR data. H.P. and J.C. analyzed the NMR data. All the authors have contributed equally towards the interpretation of the data as well as the preparation of the manuscript.

## Competing interests
The authors declare that they have no competing interests.
