## [Peer Review File · Communications Chemistry]

Reviewers' comments:

Reviewer #1 (Remarks to the Author):

In this paper authors describe molecular properties of RNA monomers in the absence or presence of PEG, which is known to be a crowding agent, on the basis of NMR analyses, DOSY and T1 measurement. Decrease of diffusion coefficient and rotational correlation time of each molecule were found upon increase of its concentration. Also, in the presence of PEG, further effect on these parameters was observed. Basically π -molecules tend to be stacking in condensed conditions, and as authors mentioned diffusion coefficients of molecules in the crowding conditions are basically decreased. Therefore, their finding includes less novelties on my view. In addition, while authors conducted good analyses of ribonucleotides properties, the manuscript is short on details and does not include a proof-of-concept which is "the effect of molecular crowding on π -stacking might result in the formation of pseudo-oligomers". I feel there is a big gap between their results and hypothesis. This paper is not suitable for publication in Communications Chemistry. I recommend this paper will be published on a journal featuring the origin of life.

Reviewer #2 (Remarks to the Author):

Review of "NMR-based analysis of nucleotide π -stacking in a crowded environment: Implications for prebiotic reactions" by Bapat et al.

This is an interesting paper, generally well presented and written. The results seem interesting, but I really can't comment as to how unexpected they are. Maybe very? That said there are several ways this paper could be improved.

First off the writing needs considerable editing for clarity and brevity. I would say the text could be pared down by ~20%. Lead-off phrases like "Significantly," "On the other hand," "Furthermore" etc. should be avoided unless they are really necessary. Words like "both" and "partly" do not need to be isolated with commas.

The mention of the LHB is unnecessary. First, it is now unclear that there was a LHB, and when it occurred, if it did, is not necessarily overlapping with when life began, which is admittedly unknown, so they are conceptually unrelated ideas. In other words, the input of ET organic materials prior to the OoL is not conceptually connected to the LHB and the LHB does not need to be mentioned.

The concept of crowding is not well explained. Is it merely high concentration? Does it matter what the co-solute is? Presumably yes, PEG being different from say ammonia. Is crowding the same as adsorption physically? This needs some careful exposition.

It was already well-known that purines self-associate more than pyrimidines, for well-understood reasons, could the authors try to explain why one would not expect that to also be true under "crowded" conditions?

It is not clear what concentration is being indicated in the X axis in Figure 1 (nucleotide or PEG?).

The meaning of Figure 2 is unclear to me. If I am reading this correctly, shouldn't the T1 decrease with decreasing temperature for smaller molecules? This does not appear to be the case from Table 1/Figure 3.

Isn't the data in Table 1 the same as that shown in Figure 3? If so, I would delete the table.

Is there anyway to be sure the decrease in T1 isn't due to other types of intermolecular interaction

besides stacking, such as hydrogen bonding or solvent organization? As mentioned on p10, H-bond mediated aggregation could be a significant contributor.

Reviewer #3 (Remarks to the Author):

The paper by Bapat, Paithankar et al. describes results of a study relevant in the context of prebiotic chemistry and related to the affinity of nucleotides to stack. The authors put the focus on the impact of additional components in prebiotic mixtures on stacking of nucleotides, which also has implications for the generation of information-bearing molecules, but was generally not considered sufficiently in studies of emergence of life.

The authors base their study on NMR of the mixtures where crowding is induced by a crowding agent, here PEG. The experimental conditions used by authors are highly relevant in the context of prebiotic chemistry. Without going in depth into the NMR analysis, their analysis technique is well chosen, I find the paper to be well written, and the study to be well conducted. I can recommend publication of the present manuscript in its current form.

Responses to the Reviewers' comments:

Reviewer #1 (Remarks to the Author):

In this paper authors describe molecular properties of RNA monomers in the absence or presence of PEG, which is known to crowd agent, on the bases of NMR analyses, DOSY and T_1 measurement. Decrease of diffusion coefficient and rotational correlation time of each molecule were found upon increase of its concentration. Also, in the presence of PEG, further effect on these parameters was observed. Basically π -molecules tend to be stacking in condensed conditions, and as authors mentioned diffusion coefficients of molecules in the crowding conditions are basically decreased. Therefore, their finding includes less novelties on my view. In addition, while authors conducted good analyses of ribonucleotides properties, the manuscript is short on details and does not include a proof-of-concept which is "the effect of molecular crowding on π -stacking might result in the formation of pseudo-oligomers". I feel there is big gap between their results and hypothesis. This paper is not suitable for publication in Communications Chemistry. I recommend this paper will be published on a journal featuring the origin of life.

Response: We thank the reviewer for their views on our manuscript. We would like to say the following in response: Although it might seem obvious that π -molecules stack, for e.g., in the context of DNA and RNA helices, aromatic amino acid side-chain intercalating in DNA/RNA, etc., it is unclear what this obviously might mean for some of the prebiotically pertinent processes where stacking could pave the way to keep the molecules together long enough, facilitating further reactions to occur. Also, it is not yet fully understood if it's the helical structure of DNA/RNA that drives the stacking, or if it's the stacking of bases, that drives the formation of helical structure. In our work, we particularly focused on discerning the differences between the stacking efficiencies of purine vs. pyrimidine mononucleotides, especially in the context of how crowding might have affected nonenzymatic templated replication. This prebiotic process is inherently prone to the complexities that arise due to the absence of precise protein machineries, thus making it all the more vulnerable to events that could affect the molecular interactions that drive them.

When nucleotides stack, they would get clustered and their effective molecular size would increase, and this is evident from both our DOSY (which probes translational diffusion process) and T_1 NMR data (which probes translational and rotational diffusion processes). Our NMR observations suggest slowing down of diffusion of nucleotides independently by DOSY and T_1 , which can be deciphered as possibly stemming from stacking of the nucleotides. This could result in the clustering of nucleotides resembling a polymer-like structure that we call pseudo-oligomers since the nucleotides are not covalently linked in these clusters. Further, this stacking tendency has been observed to increase with the nucleotide concentration, especially so for purines in the presence of crowding agents. This would essentially result in higher sequestration of purine monomers as compared to the pyrimidine monomers, affecting their availability for the prebiotically relevant reactions like

nonenzymatic template-directed replication reactions that we reported in our 2015 JME study (Bapat & Rajamani, 2015, JME). This is somewhat counterintuitive as crowding conditions tend to enhance the kinetics of reaction events (Ellis, 2001, Trends Biochem Sci). However, in our case, two of the otherwise fast reactions (under control conditions) that were originally characterized in the JME study, were the ones that slowed down most prominently. This is an observation that we could not have predicted upfront, which eventually led to our working hypothesis wherein we posited the formation of non-covalently bound oligomers under prebiotically crowded conditions. Consequently, we undertook this study, which enabled us to show that the aforesaid is possibly an important main reason for our JME study's observations.

Reviewer #2 (Remarks to the Author):

Review of "NMR-based analysis of nucleotide π -stacking in a crowded environment: Implications for prebiotic reactions" by Bapat et al.

This is an interesting paper, generally well presented and written. The results seem interesting, but I really can't comment as to how unexpected they are. Maybe very? That said there are several ways this paper could be improved.

Response: We are really grateful to the reviewer for their detailed evaluation and pertinent suggestions.

First off the writing needs considerable editing for clarity and brevity. I would say the text could be pared down by ~20%. Lead-off phrases like "Significantly," "On the other hand," "Furthermore" etc. should be avoided unless they are really necessary. Words like "both" and "partly" do not need to be isolated with commas.

Response: The text in the manuscript has been revised to reduce the wordiness wherever possible.

The mention of the LHB is unnecessary. First, it is now unclear that there was a LHB, and when it occurred, if it did, is not necessarily overlapping with when life began, which is admittedly unknown, so they are conceptually unrelated ideas. In other words, the input of ET organic materials prior to the OoL is not conceptually connected to the LHB and the LHB does not need to be mentioned.

Response: As per the reviewer's suggestion, the contextual mention of LHB has been taken out from the 'Introduction' section.

The concept of crowding is not well explained. Is it merely high concentration? Does it matter what the co-solute is? Presumably yes, PEG being different from say ammonia. Is crowding the same as adsorption physically? This needs some careful exposition.

Response: Although both crowding and adsorption result from non-specific interactions that lead to an increase in effective concentration of the reactant molecules, crowding is known to be a result of mainly repulsive background interactions, whereas attractive background interactions are what primarily lead to the adsorption of molecules on surfaces (Minton, 2006, J. Cell Sci.). Since the volume available for macromolecules differ from that for small molecules, crowding does depend on the size of the co-solute as well as the size of the target molecule being studied. Thus, polymers like PEG and dextran can lead to more crowding than say a co-solute like ammonia. Additionally, a bigger sized reactant molecule, for e.g. stacks of nucleotides, would potentially experience larger steric repulsion from background molecules than individual nucleotide monomers, thus making it difficult for the bigger molecule to diffuse through the crowded medium. Towards demonstrating the generality of our observations, we have included data obtained for reactions wherein dextran was used as a crowding agent. Overall, the trends observed were similar for reactions where PEG and dextran were used to simulate crowded conditions.

It was already well-known that purines self-associate more than pyrimidines, for well-understood reasons, could the authors try to explain why one would not expect that to also be true under "crowded" conditions?

Response: Molecular crowding, in general, tends to aggregate the molecules due to volume exclusion effects. Hence, such phenomenon might also affect the stacking properties of the nucleotides. In addition to what we have elaborated upon in our response to a similar comment made by Reviewer 1, we would like to add the following: Due to volume exclusion effects, the pyrimidines might also stack to a greater extent than when in plain buffered solution. However, to the best of our knowledge, the extent of stacking of ribonucleotides has not been analyzed in the presence of crowding agents before. We were particularly interested in knowing whether the effect of molecular crowding on stacking would be different for purines vs. pyrimidines. Although one might expect similar stacking tendencies in the presence of crowding agents, its extent might vary, which is what our results indicate. This has implications for how the sequestration of the nucleotides within these stacks might vary, which would subsequently affect the availability of these solutes to the reactions in question.

It is not clear what concentration is being indicated in the X axis in Figure 1 (nucleotide or PEG?).

Response: The X-axis in Figure 1 indicates 'Nucleotide concentration (in mM).' Appropriate changes have been made to reflect this in the figure.

The meaning of Figure 2 is unclear to me. If I am reading this correctly, shouldn't the T_1 decrease with decreasing temperature for smaller molecules? This does not appear to be the case from Table 1/Figure 3.

Response: Yes, Figure 2 shows that T_1 decreases with a decrease in temperature for small molecules (having tau-c values to the left of T_1 minima in the plot). As shown in Figure 3 and Supplementary Table 2 in the revised version, we observed a decrease in T_1 values when going from higher (25 °C) to lower temperature (10 °C). The only exception for this observation is the single data point for 10 mM GMP in the presence of PEG, which has been reasoned to form G-quadruplex at higher nucleotide concentrations. This was also confirmed by the presence of NH peak in ^1H NMR spectrum at a position characteristic of G-quadruplex. Based on our experimental observations for the T_1 values for nucleotides, in the presence and absence of the crowding agent, we have concluded that the nucleotides/nucleotide clusters still act as small molecules even in the presence of the crowding agents (except for 10 mM GMP, in the presence of 18% PEG 8000). Figure 3 also highlights that with an increase in nucleotide concentration, the difference between the T_1 values of hot and cold samples is reduced suggesting that the tau-c is increasing and molecule is moving closer to T_1 minimum in the plot. This increase in tau-c with an increase in nucleotide concentration is interpreted as an overall increase in molecular size.

Isn't the data in Table 1 the same as that shown in Figure 3? If so, I would delete the table.

Response: The Table 1 has been moved to the supplementary information section as Supplementary Table 2. We have chosen to retain the table as it would make the actual numbers available for interested readers.

Is there anyway to be sure the decrease in T_1 isn't due to other types of intermolecular interaction besides stacking, such as hydrogen bonding or solvent organization? As mentioned on p10, H-bond mediated aggregation could be a significant contributor.

Response: The decrease in T_1 suggests slowing down of motion (both translational and rotational) and thus having a more efficient spin-lattice relaxation process. This could

happen due to multiple reasons as mentioned by the reviewer. As for hydrogen bonding, the decrease in T_1 with an increase in the concentration would not be gradual, as once two nucleotides (or four in case of the formation of a tetrad) are H-bonded, the H-bonding sites saturate. Stacking can, however, continue until the persistence length of the stacked structure allows it to go, and keeps on decreasing the T_1 gradually. Nonetheless, the possibility of H-bonding (in addition to stacking) may not be ruled out completely. The discussion has been modified to include this possibility.

Reviewer #3 (Remarks to the Author):

The paper by Bapat, Paithankar et al. describes results of a study relevant in the context of prebiotic chemistry and related to the affinity of nucleotides to stack. The authors put the focus on the impact of additional components in prebiotic mixtures on stacking of nucleotides, which also has implications for the generation of information-bearing molecules, but was generally not considered sufficiently in studies of emergence of life.

The authors base their study on NMR of the mixtures where crowding is induced by a crowding agent, here PEG. The experimental conditions used by authors are highly relevant in the context of prebiotic chemistry. Without going in depth into the NMR analysis, their analysis technique is well chosen, I find the paper to be well written, and the study to be well conducted. I can recommend publication of the present manuscript in its current form.

Response: We thank the reviewer for evaluating our study and for their kind words. We are particularly glad to note that the reviewer found this study insightful, especially in the context of emergence of life on prebiotic Earth.

Reviewers' comments:

Reviewer #1 (Remarks to the Author):

I understand the author's research interest. Authors mentioned in the response, *' it is not yet fully understood if it's the helical structure of DNA/RNA that drives the stacking, or if it's the stacking of bases, that drives the formation of helical structure. '* and *'This (T1 and DOSY data) could result in the clustering of nucleotides resembling a polymer-like structure that we call pseudo-oligomers since the nucleotides are not covalently linked in these clusters.'* In the manuscript, size and structure of pseudo-oligomer formed by nucleotides in the crowded environment were not well studied. To improve this paper, additional supporting data, for example, 1) direct evidence of nucleotide-nucleotide interaction and 2) alternative studies investigating molecular size and architecture of clustering of nucleotides are required. Such data will be great help for readers including unfamiliar with prebiotic chemistry field.

Reviewer #2 (Remarks to the Author):

The authors have adequately addressed my comments from the previously submitted version.

"NMR-based analysis of nucleotide π -stacking in a crowded environment: Implications for prebiotic reactions"

Reviewers' comments:

Reviewer #1 (Remarks to the Author):

I understand the author's research interest. Authors mentioned in the response, '*it is not yet fully understood if it's the helical structure of DNA/RNA that drives the stacking, or if it's the stacking of bases, that drives the formation of helical structure.*' and '*This (T1 and DOSY data) could result in the clustering of nucleotides resembling a polymer-like structure that we call pseudo-oligomers since the nucleotides are not covalently linked in these clusters.*' In the manuscript, size and structure of pseudo-oligomer formed by nucleotides in the crowded environment were not well studied. To improve this paper, additional supporting data, for example, 1) direct evidence of nucleotide-nucleotide interaction and 2) alternative studies investigating molecular size and architecture of clustering of nucleotides are required. Such data will be great help for readers including unfamiliar with prebiotic chemistry field.

We thank Reviewer #1 for their critical evaluation of the revised manuscript. At the outset, we do want to assure the Reviewer that we are very much on the same page with regards to acquiring and collating the data that they have alluded to in points 1 and 2 of their comment. And, we are also in agreement with their following statement that "Such data will be great help for readers including unfamiliar with prebiotic chemistry field.". On behalf of us all, I would like to assure that discerning the aforementioned aspects in detail is of utmost priority for us. The interaction that we proposed in the manuscript is based on diffusion coefficient and transverse relaxation time data, which reflects on the potential stacking interactions between the nucleotides. Intrinsically, this is a difficult interaction to probe by mass determining techniques as it would require the usage of a particular concentration of the samples and the running of these samples through columns, both of which would adversely affect the stacks that are very delicate to begin with (which exist without any support from covalent or other non-covalent interactions). The two orthogonal approaches in NMR that we used, diffusion coefficient analysis (probed by diffusion optimised spectroscopy), and rotational and translation diffusion (probed by longitudinal relaxation time), gave a strong indication of large soluble species in solution. As of now, the one thing that can be said is that both H-bonding and stacking interactions could be involved in the decrease seen, especially, in T1 studies and these might be in addition to a possible contribution from solvent organization. We have detailed our response in this regard in the previous submission cycle (last response to Reviewer #2's comment**), and the discussion was modified in that revised version to include the aforementioned possibilities. Further characterisation of the structure of the species and the detailed characterization of the underlying mechanism(s) that drive such physico-chemical events is, therefore, very much on cards. Nonetheless, this is a full project unto itself and will have to be undertaken in a very systematic manner before we can actually comprehensively discern the molecular details of the potential nucleotide clusters.

Having said the above, we believe that this is currently beyond the scope of this manuscript. The work detailed in here is about the effect of co-solutes on the nonenzymatic oligomerization and

the detection of oligomer sized entities in such scenarios. Thus, it is an important demonstration of how heterogeneity in the prebiotic soup (even to a relatively small extent), actually would impinge on prebiotically pertinent nonenzymatic processes; something that has also been reflected in just a handful of other studies to date. Significantly, both DOSY and T1 relaxation studies, which are totally different techniques, ended up giving similar results highlighting our primary observation. In conclusion, we sincerely hope that the reviewer appreciates the above mentioned aspects and sees the merit in this work being considered for publication without further delay.

Reviewer #2 (Remarks to the Author):

The authors have adequately addressed my comments from the previously submitted version.

We are grateful to Reviewer #2 for confirming that we have indeed addressed their valuable comments from the first round of the review process. Thank you.

Reference:

****Referee #2's comment:** Is there anyway to be sure the decrease in T1 isn't due to other types of intermolecular interaction besides stacking, such as hydrogen bonding or solvent organization? As mentioned on p10, H-bond mediated aggregation could be a significant contributor.

Response: The decrease in T1 suggests slowing down of motion (both translational and rotational) and thus having a more efficient spin-lattice relaxation process. This could happen due to multiple reasons as mentioned by the reviewer. As for hydrogen bonding, the decrease in T1 with an increase in the concentration would not be gradual, as once two nucleotides (or four in case of the formation of a tetrad) are H-bonded, the H-bonding sites saturate. Stacking can, however, continue until the persistence length of the stacked structure allows it to go, and keeps on decreasing the T1 gradually. Nonetheless, the possibility of H-bonding (in addition to stacking) may not be ruled out completely. The discussion has been modified to include this possibility.